# Roles of Non-Coding RNAs on Anaplastic Thyroid Carcinomas

**DOI:** 10.3390/cancers12113159

**Published:** 2020-10-28

**Authors:** Plabon Kumar Das, Saharia Yeasmin Asha, Ichiro Abe, Farhadul Islam, Alfred K. Lam

**Affiliations:** 1Department of Biochemistry and Molecular Biology, University of Rajshahi, Rajshahi 6205, Bangladesh; plabonsb5208@gmail.com (P.K.D.); saharia.yeasmin@gmail.com (S.Y.A.); 2School of Medicine, Griffith University, Gold Coast, QLD 4222, Australia; i.abe@griffith.edu.au; 3Department of Endocrinology and Diabetes Mellitus, Fukuoka University Chikushi Hospital, Chikushino, Fukuoka 818-8502, Japan; 4Institute for Glycomics, Griffith University, Gold Coast Campus, Gold Coast, QLD 4222, Australia

**Keywords:** Anaplastic thyroid carcinoma, non-coding RNAs, miRNAs, lncRNAs, siRNAs, targeted therapy

## Abstract

**Simple Summary:**

Anaplastic thyroid carcinoma (ATC) is a biological aggressive human carcinoma and causes most of the thyroid cancer related deaths. Non-protein producing RNAs, such as long non-coding RNAs (lncRNAs) and microRNAs (miRNAs), play crucial roles in formation and therapy resistance of ATC by regulating the production of key proteins. These RNAs alter the proteome of cells, which in turn controls cancer cells’ growth, division, invasion, migration, metastasis, and recurrence of ATC. Thus, the exogenous manipulation of these RNAs could interfere cells growth and development of ATC. Considering this, we discuss the roles of various miRNAs and lncRNAs in the pathogenesis of ATC and their potential therapeutic applications.

**Abstract:**

Anaplastic thyroid cancer (ATC) remains as one of the most aggressive human carcinomas with poor survival rates in patients with the cancer despite therapeutic interventions. Novel targeted and personalized therapies could solve the puzzle of poor survival rates of patients with ATC. In this review, we discuss the role of non-coding RNAs in the regulation of gene expression in ATC as well as how the changes in their expression could potentially reshape the characteristics of ATCs. A broad range of miRNA, such as miR-205, miR-19a, miR-17-3p and miR-17-5p, miR-618, miR-20a, miR-155, etc., have abnormal expressions in ATC tissues and cells when compared to those of non-neoplastic thyroid tissues and cells. Moreover, lncRNAs, such as H19, Human leukocyte antigen (HLA) complex P5 (HCP5), Urothelial carcinoma-associated 1 (UCA1), Nuclear paraspeckle assembly transcript 1 (NEAT1), etc., participate in transcription and post-transcriptional regulation of gene expression in ATC cells. Dysregulations of these non-coding RNAs were associated with development and progression of ATC by modulating the functions of oncogenes during tumour progression. Thus, restoration of the abnormal expression of these miRNAs and lncRNAs may serve as promising ways to treat the patients with ATC. In addition, siRNA mediated inhibition of several oncogenes may act as a potential option against ATC. Thus, non-coding RNAs can be useful as prognostic biomarkers and potential therapeutic targets for the better management of patients with ATC.

## 1. Introduction

Anaplastic thyroid carcinoma (ATC) accounts for 1–2% of all thyroid cancers, however, it is responsible for the most cancer deaths caused by thyroid cancer despite therapeutic intervention [1,2]. ATC is one of the most aggressive solid tumours, which contributes to 3.6% of all human cancers. Many patients with ATC die within a year of diagnosis [3,4,5,6]. Several cellular, genetic, and epigenetic aberrations have been reported to orchestrate the tumourigenesis of ATC [7,8]. These aberrations lead to the dysregulation of cellular signalling which as a result enhances cellular proliferation and accelerates tumourigenesis. In contrast to differentiated thyroid carcinoma such as papillary thyroid carcinoma and follicular thyroid carcinoma cells, cells in ATC do not normally retain biological features or functions of follicular thyroid cells [9,10]. ATC is characterized by aggressive local progression as well as high rates of distant metastasis, which lead to rapidly fatal clinical outcomes [1,2,9,10]. The treatment option of the patients with ATC includes surgery, radiotherapy, and/or chemotherapy (multimodal therapy), targeted therapy, and immunotherapy [1]. Among them, molecular targeted therapies are the promising emerging treatment modalities [7]. Hence, the management of ATC is always difficult; therefore, exploring new therapeutic modalities may provide favourable prediction as well as increasing the overall survival rate of patients [11]. Recent studies also revealed more information about genetic profiles on ATC and these genomic data are useful for the future advances of targeted therapy [12,13,14].

MicroRNAs (miRNA) regulate gene expression and therefore protein production by binding to their target mRNAs. miRNAs play crucial role in the regulation of many cellular processes such as growth, development, metabolism, proliferation and apoptosis etc. [15]. Many miRNAs are abnormally expressed in thyroid carcinomas. These dysregulations in miRNA expression may contribute to the pathophysiology of tumourigenesis and disease progression in ATC [15]. Similarly, long non-coding RNAs (lncRNAs) also play critical role in ATC by regulating transcription and epigenetic or post-transcriptional regulation of gene expression [16]. They have been found to be implicated in epithelial to mesenchymal transition (EMT) in ATC, which enhances the metastatic potential of ATC. For example, suppressing the expression of lncRNA metastasis-associated lung adenocarcinoma transcript 1 (MALAT1) in ATC cells halted cell proliferation, migration and invasion. In addition, inhibition of MALAT1 caused increased apoptosis and autophagy formation [16].

In this review, we describe how non-coding RNAs (ncRNAs) function in epigenetic regulation of normal cells and ATC cells. Furthermore, the roles of specific miRNAs, lncRNA, and siRNA molecules in ATC as well as the potential use of ncRNAs in designing targeted therapy against ATC are also illustrated.

## 2. Non-Coding RNAs as the Important Players in the Regulation of Gene Expression

MicroRNAs (miRNAs/miRs), one of the classes of small non-coding RNAs comprising 19–25 nucleotides, mediate post-transcriptional gene silencing by partial complementarity with the 3′untranslated region (3ʹ UTR) of the target mRNAs (Figure 1) [17]. miRNAs are one of the most vigorously studied subclasses of non-coding RNAs as they play a regulatory role in epigenetic control of cells [18,19]. miRNA mediated gene silencing occurs through endonucleolytic cleavage or by translational repression [20,21]. Small interfering RNAs (siRNAs), another member of non-coding RNA family, have some similarities with miRNAs as far as the length and the pathways of their biogenesis are concerned. miRNAs are excised from stem-loop structures, whereas siRNAs are derived from long entirely complementary double-stranded RNA precursors primarily of exogenous origins, such as retrotransposons and viral sequences [21,22]. Both siRNAs and miRNAs are associated with translational repression and mRNA cleavage, which mainly depends on their homology to their mRNA targets (Figure 1).

Long non-coding RNAs (lncRNAs) are another type of non-coding RNAs that control transcription, translation, and mRNA stability in the processes of cell differentiation and development [23,24,25]. The tissue specific expression of lncRNAs is lower than the protein coding genes. They are generated via pathways similar to that of protein coding genes, with similar histone modification and splicing pattern. However, lncRNAs are typically without having an open reading frame (ORF) and found in the chromatin inside the nucleus [26]. PIWI-interacting RNAs (piRNAs), like other non-coding RNAs, have also been found to contribute in epigenetic programming [27,28]. A miRNA-like post-transcriptional gene silencing is mediated by piRNA through transposon sequences overlapping with the 3′ or 5′UTRs of mRNAs [29,30]. Though piRNAs show their expressions in somatic tissue, they are mainly transcribed in germline cells. In somatic tissue, the majority of piRNAs are transcribed in the known transcript, whereas they are encoded as piRNA clusters in germline cells [31]. In contrast to lncRNAs and miRNAs, piRNAs direct transposon cleavage, in order to protect the genome against transposon-induced insertional mutagenesis to maintain genome integrity [32]. In this review, we will focus only on miRNAs, lncRNAs, and siRNAs as they are mainly involved in pathogenesis in ATC.

## 3. Roles of miRNAs in Anaplastic Thyroid Carcinoma

MiRNA-mediated gene silencing in tumourigenesis has been a prominent topic over the last few decades due to a variety of reasons. First, various cellular processes including cell differentiation, division, proliferation, and apoptosis are controlled by miRNA expressions. Second, most of the miRNAs are in or near the fragile tumour-related genomic regions, and at the amplification or broken point region. The locations of these genes at these fragile regions could attribute to their aberrant expression by inducing genomic instability in various cancers [33]. Third, abnormal expressions of miRNAs have been noted in a variety of tumour tissues and cancer cells. Finally, alterations in the sequence of the miRNA precursors affect the processing of miRNA maturing, which in turn results in cancer susceptibility [34]. This information suggest that miRNAs are closely related to tumourigenesis in human cancers. Therefore, selective manipulation of dysregulated miRNAs expression could inhibit the formation of tumour [35]. In ATC, miRNAs are associated with the development, metastasis, and prognosis of patients with the cancer. Thus, altering the expression of these miRNA could be a potential therapeutic option against patients with ATC [36,37,38].

Several miRNAs have been found to be involved in identifying clinical aggressiveness of ATC and prognosis of patients with ATC (Table 1) (Figure 2). For example, one of the very commonly dysregulated miRNAs in ATC is miRNA-205-5p, which plays critical roles in angiogenesis and EMT [37]. Notably, miRNA-205 (miR-205) contributes to tumour vascularization and tumour invasion through regulating the expression of *VEGF-A* and *ZEB1* genes in melanoma, glioblastoma, ovarian carcinoma, and breast carcinoma. Two ATC cell lines (MB-1, BHT-101) were transfected with the miR-205 expression vector and significant inhibition of angiogenesis and EMT was detected through suppression of *VEGF-A*, *ZEB1*. In addition, miR-205 overexpression impeded the migration, invasion, and tube formation of endothelial cells. Furthermore, in vivo miR-205 expression resulted in inhibition of tumour growth, vascularization and invasion [37]. Thus, miR-205 expression could be used as potential option in order to halt or inhibit both angiogenesis and EMT in ATC.

miR-19a is involved in the progression of ATC [38]. A significant reduction of cell proliferation was noticed in miR-19a downregulated ATC cells (8505c). Robust reduction in cell growth and viability after miR-19a inhibition was reported due to the induction of apoptosis by activation of caspases (caspase 9). *PTEN,* a tumour suppressor and is one of the known targets of miR-19a. Exogenous inhibition of miR-19a could activate PTEN in ATC cells which subsequently leads to reduced proliferation and decreased cell growth. Moreover, PTEN inactivation could correlate with the progression and aggressiveness of ATC [39]. Thus, the activation of PTEN by modulating miR-19a could provide better management of patients with ATC.

A study determining the role of miR-17-92 cluster suggested that miR-17-3p and miR-17-5p, two of the RNAs in miR-17-92 cluster, were overexpressed in ATCs when compared to adjacent non-cancerous tissues [40]. To determine the role of miR-17-3p and miR-17-5p, ATC cells (ARO and FRO) were transfected with antisense oligonucleotides containing locked nucleic acids, inhibitors of these miRNAs. Inhibition of miR-17-3p expression resulted in complete growth arrest followed by apoptosis of ATC cells. Whereas, miR-17-5p inhibitor induced substantial growth reduction and led to cellular senescence in ATC cells. However, inhibition of miR-18a, another member of the cluster, moderately attenuated the cell growth which indicated functional differences among the members of the cluster in ATC cells [40]. Inhibition of miR-17-3p expression resulted in increased apoptosis by caspase activation. However, transfection with other inhibitors of miR-17-3p cluster did not cause caspase activation. Thus, miR-17-3p could have a distinct function from other members of the cluster [40]. Thus, further studies are required to identify target mRNA for miR-17-3p in ATC.

X-linked inhibitor of apoptosis protein (XIAP) has been found to contribute to cancer growth and progression [41]. Inhibition of XIAP induced apoptosis in ATC. A study comparing miRNA mediated regulation of XIAP in non-neoplastic thyroid cells (Nthy-ori 3-1) and ATC cells (8305C) found that miR-618 targets XIAP expression, which results in inhibition of tumour growth [41]. In addition, ectopic miR-618 expression was associated with the prevention of invasion and migration of ATC (8305C) cells by silencing *XIAP* expression [41]. Therefore, the over-expression of miR-618 could inhibit ATC cells by targeting *XIAP* gene.

miR-20a is another example of miRNA inhibiting the growth of ATC cells by regulating the *LIMK1* expression [42]. LIMK1 plays critical role in tumour cell invasion and metastasis in various cancers [43,44,45,46]. LIMK1 overexpression could increase the invasiveness of breast and prostate cancer cells both in vitro and in vivo, and knockdown of *LIMK1* could diminish invasion of breast carcinoma and prostate carcinoma cells [46,47]. Rho signalling regulates the activity of LIMK1, which modulates actin dynamics by controlling the activity of the cofilin family proteins [48,49]. miR-20a expression was significantly higher in ATC tissues when compared with differentiated thyroid carcinoma, benign thyroid lesions, and non-cancerous thyroid tissues. In addition, the overexpression of miR-20a in ATC cells (C643) caused reduced proliferation and tumour growth both in vitro and in vivo by inducing silencing of LIMK1 expression [42]. Also, knockdown of LIMK1 resulted in inhibition of cellular invasion and migration of ATC cells [42].

Contrary to miR-20a, miR-155 could stimulate the growth of ATC cells by regulating the expression of suppressor of cytokine signalling 1 (SOCS1) [50]. miR-155 inhibits apoptosis as well as promotes proliferation, invasion and migration of ATC (8305c cells and FRO cells) [50]. SOCS1 is a member of the suppressor of cytokine signalling (SOCS) family, which negatively regulates cytokine signal transduction [51]. Aberrant expressions of miR-155 and *SOCS1* showed inverse correlations in ATC. It was found that miR-155 lowers SOCS1 levels by binding to *SOCS1* 3′-UTR, therefore promoting its degradation [50]. In addition, miR-155 expression was associated with cervical metastasis and extra-thyroidal invasion in ATC. miR-155 inhibition caused a significant increase in the expression level of *SOCS1*. Thus, the expression of *SOCS1* reversed the effects of miR-155 suppression on cell proliferation, apoptosis as well as invasion. Furthermore, miR-155 in association with downregulation of SOCS1 has been implicated in the progression of lung carcinoma and anaplastic large cell lymphoma [52,53]. Therefore, miR-155 could be a potential target for therapeutics against patients with various cancers, including ATC.

miR-125b is downregulated and identified as a tumour suppressor in various cancers including thyroid cancer [54]. miR-125b expression was downregulated in ATC tissues and cells when compared to that of adjacent non-neoplastic thyroid tissues and non-neoplastic thyroid follicular cells (Nthy-ori 3-1), respectively. There was an inverse correlation between phosphoinositide 3-kinase catalytic subunit delta (PIK3CD) and miR-125b expression noted in ATC tissues and cells (SW1736 and 8305C) [54]. PIK3CD contributes in the PI3K signalling pathway, which plays an important role in the development of various cancers [55,56]. A negative relationship between miR-125b and PIK3CD was noted, as miR-125b targets and inhibits the expression level of PIK3CD in ATC tissues and cells. Ectopic miR-125b expression inhibited the migration and invasion of ATC cells whereas PIK3CD overexpression reversed this effect. Furthermore, exogenous miR-125b expression in ATC cells (SW1736 and 8305C) suppressed PI3K, phospho-Akt, and phospho-mTOR expression [54]. Thus, suppression of cell migration and invasion by miR-125b expression through inhibiting PIK3CD suggested a potential therapeutic option for the treatment of patients with ATC.

Expression of miR-483 was increased, while the expression of partitioning-defective 3 (Pard3) was decreased in ATC cells and tissues [57]. A member of Pard family namely Pard 3, controls cellular polarity, migration, and cell division [58]. The study also investigated the role of miR-483 and Pard3 on the TGF-β 1-mediated progression of ATC. TGF-β1 was found to regulate miRNA driven migration and tumourigenesis in various cancers [59,60]. In ATC, activated TGF-β1 recruits Pard3 in order to control cell polarity and promote cell migration [57]. Moreover, inhibition of miR-483 expression increased Pard3 expression, thereby promoting TGF-β1-induced cell migration and invasion. It was shown that after binding to Pard 3, miR-483 targets and binds to Pard 3 to inhibit its expression in ATC cells which as a result induces TGF-β1-mediated cell proliferation, migration, and invasion, both in vitro and in vivo [57]. Clinically, poor disease-free survival rates were observed upon downregulation of miR-483 and upregulation of Pard3 in patients with ATC [57]. Furthermore, it was proven that overexpression of Pard3 inhibits TGF-β1 mediated EMT and Tiam1-Rac signalling in ATC cells [57].

In comparison to non-neoplastic thyroid tissues and differentiated thyroid carcinomas, the expression of miR-25 and miR-30d was significantly downregulated in ATC tissues [61]. Importantly, ectopic expression of miR-25 and miR-30d inhibited proliferation and colony formation of ATC cells by arresting cells at G2/M phase. Mechanistically, miR-25 and miR-30d reduced cancer cells growth and proliferation by targeting the expression of oncogene, polycomb protein enhancer of zeste 2 (*EZH2*) [62]. *EZH2* is one of the oncogenes upregulated in ATCs when compared to differentiated thyroid carcinomas. This inverse relationship between miRNAs and oncogene *EZH2* indicates an active role of these miRNAs in the pathogenesis of ATC.

MiR-34b was downregulated in cells derived from metastatic ATC (BHT-101) compared to the cells derived from primary ATC (8505C) [63]. Exogenous overexpression of miR-34b in ATC cells showed decreased cell proliferation, decreased wound healing, reduced cell cycle progression and increased apoptosis [63]. Besides, liposome-mediated delivery of miR-34 resulted in downregulation of vascular endothelial growth factor-A (VEGF-A) in ATC cells. VEGF-A is a principle factor for angiogenesis in cancer [63]. In vivo experiments suggested that intravenous administration of liposome-loaded miR-34b reduced the size of tumour formation when compared to control [63]. A promising tumour-suppressing role of miR-34b via VEGF-A regulation in ATC cells was demonstrated using both in vitro and in vivo model by the study [63]. Thus, miR-34b could be a useful therapeutic strategy, in addition to the VEGF-A, for design of targeting therapy in patients with ATC.

The epidermal growth factor receptor (EGFR) is overexpressed in ATC cells, which dictates its crucial role in tumourigenesis of ATC [64]. Epidermal growth factor (EGF) starts a signalling cascade after binding to EGFR, and this signalling may enhance migration and invasiveness of ATC cells. Importantly, there is an interaction identified among miR-200 family members, EGF/EGFR and EMT process in ATC cells. Induction of EGF in ATC cells leads to the loss of miR-200 expression and increases Rho/ROCK activity, resulting in increased EMT and subsequent cancer invasion [64]. Increased vimentin expression was noted in non-neoplastic human thyroid cells (Nthy-ori 3-1) followed by EGF treatment. On the other hand, EGFR silencing results in vimentin downregulation in ATC cells (SW1736), which indicates that EGF induces EMT in non-neoplastic thyroid cells, while EGFR silencing in ATC cells alters EMT. Moreover, EGF treatment downregulates miR-200s in non-neoplastic thyroid cells, whereas EGFR silencing increases miR-200s expression in ATC cells, indicating miR-200 family and EGF/EGFR mediated regulation of EMT [64]. Furthermore, Rho/ROCK signalling is crucial in TGFβ-induced EMT as inhibition of the Rho effector (ROCK) results in inhibition of TGFβ-induced EMT [65,66]. The study found an increase of Rho/ROCK activity in EGF-treated non-neoplastic thyroid cells and the reverse was noted in EGFR-silenced ATC cells [64]. Therefore, a model was proposed for EMT regulation by EGF/EGFR, miR-200s, Rho/ROCK and EMT markers suggesting that miR-200 upregulation could serve as a potential therapeutic option for ATC. Other microRNAs like miR-146a/b was found to be upregulated in ATC, whereas microRNA-30 family and let-7 family were downregulated [67,68,69]. Alterations of these microRNA expressions caused a significant change in cell proliferation, cell growth, and cell differentiation in ATC cells. It was found that aforementioned miRNAs modulate their actions via affecting various molecules, such as PTEN (microRNA-146a/b), p53 (microRNA-146a/b), Beclin1 (autophagy-promoting protein; microRNA-30 family) and RAS (let-7 family) [67,68,69]. On the other hand, upregulated microRNA-195 inhibited progression of ATC by affecting the expression of VEGF and p53 [70]. Moreover, another upregulated miRNA miR-126 also reported to be reduced tumour growth by suppressing angiogenesis via reduction of VEGF expression and increasing apoptosis in ATC [71]. Hence, studies of the roles of microRNA in anaplastic thyroid carcinoma could provide potential useful information for target therapeutic approaches on anaplastic thyroid carcinoma.

## 4. Roles of lncRNAs in Anaplastic Thyroid Carcinoma

Abnormal expressions of lncRNAs have been implicated in the tumuorigenesis, progression, and chemoresistance of different human cancers, including ATCs (Table 2) (Figure 2) [72,73]. For example, *H19* gene encodes 2.3 kb long lncRNA, higher expression of this lncRNA stimulates initiation and progression and therapy resistance of ATC [73,74]. Another study reported H19 lncRNA mediated cancer initiation, progression and metastasis [75], indicating that H19 lncRNA could be used as a potential tumour marker for ATC. Moreover, inhibition of H19 expression by siRNA resulted in significantly decreased proliferation, invasion, as well as increased apoptosis of ATC cancer cells (8505C) [74]. Distant lung metastasis represents one of the most lethal threats of thyroid carcinomas and H19 has been proven to be engaged with lung metastasis in other tumour types [75]. An experimental metastasis model using a murine model of ATC unveils the functions of H19 lncRNA mediated tumour growth and metastasis [74]. This model used bioluminescence imaging, which has been a well-established method to investigate in vivo metastasis and tumour growth [76,77]. The study suggested that their bioluminescence measurements was consistent with tumour volumes during necropsy, despite of the fact that tumour bioluminescence could be affected by targeted agent mediated tumour necrosis and alterations in tumour vascularity. Targeting H19 lncRNA using shRNA reduces H19 expression level in vivo, which subsequently resulted in a strong reduction of tumour burden and inhibition of metastases in an orthotopic ATC mouse model [74]. Furthermore, targeting H19 inhibited tumour metastases by approximately eight-fold in comparison to that of control [74]. Thus, H19 might be a potential target for molecular therapy for the patients with ATC.

HLA complex P5 (HCP5) another lncRNA has been implicated in the pathogenesis of follicular thyroid carcinoma and anaplastic thyroid carcinoma by regulating the expression of tumour suppressor miR-128-3p [78]. HCP5 expression was significantly increased in ATC cells and tissue samples when compared to that of non-neoplastic human thyroid follicular cells (Nthy-ori 3-1) and adjacent non-neoplastic thyroid tissues, respectively [78]. Moreover, knockdown of HCP5 caused reduced cell viability, while elevated apoptotic rate and caspase -3/7 activity in ATC cells (ARO and SW1736). It was demonstrated that miR-128-3p was the target gene of HCP5. Thus, the higher expression of HCP5 lncRNA in ATC cells and tissues caused a significant reduction of miR-128-3p in ATC cells and tissues. Overexpression of miR-128-3p by suppressing HCP5 resulted in significant reduction of ATC cell viability and induction of apoptosis [78]. However, these effects of miR-128-3p on ATC cell viability and apoptosis were reversed when the inhibitor of miR-128-3p was applied [78]. This result could lead to a potential approach via regulating miR-128-3p for the treatment of patients with ATC.

The underlying molecular mechanism of lncRNA Urothelial carcinoma-associated 1 (UCA1) mediated progression of ATC has been described [79]. The expression levels of UCA1 were significantly upregulated in ATC cells (SW1736 and KAT-18) and tissues in comparison to that of non-neoplastic human thyroid cells (Nthy-ori3-1) and adjacent non-neoplastic tissues, respectively [79]. Suppression of UCA1 induced a significant reduction of ATC cells viability, proliferation, migration and invasion in vitro and inhibited tumour growth in vivo by repressing the expression of *c-myc* proto-oncogene [79]. miR-135a could directly bind to the 3’UTR of *c-myc* and UCA1 may bind to miR-135a. Thus, UCA1 represses the binding of c-myc with miR-135a, which inhibits miR-135a expression and increases the expression of c-myc. *c-myc* oncogene is overexpressed in many human cancers, including ATC and plays crucial role in the regulation of several cellular processes, including protein synthesis, cell growth and proliferation in various cell types [80,81]. Therefore, due to its oncogenic role in promoting ATC cell proliferation, UCA1 could be used as a potential target for treatment of ATC.

Nuclear paraspeckle assembly transcript 1 (NEAT1) is a lncRNA with a length of 4-kb and localized to the nucleus [82]. *NEAT1* acts as an oncogene and involved in the pathogenesis of various human cancers [83,84,85,86]. It plays critical roles in cancer chemoresistance several cancer types such as in osteosarcoma, ovarian carcinoma, leukaemia, and ATC [82,87,88,89]. In ATC, NEAT1 was upregulated in tissues and cells (SW1736 and 8505C) when compared to that of non-neoplastic human thyroid tissues and non-neoplastic thyroid cells (Nthy-ori 3-1), respectively. Silencing of NEAT1 resulted in decreased resistance to the chemotherapeutic reagent, cisplatin of ATC cells (SW1736 and 8505C). In addition, NEAT1 binds to miR-9-5p to suppress its expression, while it was also suggested that miR-9-5p directly target sperm-associated antigen 9 (SPAG9). Thus, NEAT1 silencing lead the overexpression of miR-9-5p, which in turn sensitized ATC cells to cisplatin by the suppression of SPAG9 expression [82]. Therefore, NEAT1 should be considered as a potential therapeutic option as silencing of NEAT1 decreased cisplatin -resistance by reducing miR-9-5p sponging and regulating SPAG9 expression.

To add to the roles of lncRNAs in ATC, another lncRNA Prader Willi/AngelmanRegion RNA5 (*PAR5*), which is downregulated in ATC, acts as a tumour suppressor by repressing Enhancer of Zeste Homolog 2 (EZH2) activity [90]. *PAR5* overexpression in ATC cells (FRO and 8505c) resulted in reduced cell proliferation and cell migration ability, which indicated the anti-oncogenic role played by this lncRNA and its downregulation could lead to development of ATC. *PAR5* interacts with EZH2 and reduces its protein level in thyroid cancer cells [90]. Moreover, *PAR5* escapes E-cadherin from the negative regulation of EZH2 by binding to *EZH2* promoter and inhibiting its expression [90]. Recent studies suggested EZH2 mediated inhibition of ATC cell differentiation, which paved the way to investigate the molecular mechanism regarding the effects of *PAR5* on EZH2 expression [91]. EZH2 was over expressed in ATCs and it inhibited thyroid cell differentiation [91]. Moreover, it was demonstrated that *PAR5* directly binds EZH2 and restoration of *PAR5* reduced EZH2 protein and H3K27me3 (DNA packaging protein Histone 3) levels in the absence *of* changes in *EZH2* mRNA level, which suggested a consistency with some previous reports about *PAR5* mediated effect on EZH2 stability [92,93]. Consistently, the study further found a negative correlation between EZH2 protein and *PAR5* RNA levels, strongly supporting the effects of *PAR5* on EZH2 [91]. As lack of E-cadherin expression is a hallmark of EMT [94,95,96,97], EZH2 could repress E-cadherin expression through H3K27 tri-methylation at its promoter [98]. ATC-*PAR5* overexpressing cells were seen to be high in E-cadherin protein levels, as *PAR5* restoration inhibited the binding of EZH2 to E-cadherin promoter, which subsequently reduces EZH2 mediated inhibitory activity on E-cadherin expression [90]. Therefore, it was cleared that *PAR5* exhibits its anti-tumour activity by impairing the oncogenic roles of *EZH2*.

LncRNA *PVT1* modulates thyroid cancer cells’ proliferation by recruiting EZH2 and regulating thyroid-stimulating hormone receptor [99]. The gene that encodes lncRNA *PVT1* resides in the well-known cancer-risk region of chromosome 8q24 [100]. Accordingly, lncRNA *PVT1* plays pivotal roles in a variety of human cancers, such as ovarian carcinoma, pancreatic carcinoma, and non-small cell lung carcinoma [101,102]. Moreover, LncRNA *PVT1* was significantly upregulated in ATC tissues, as well as in ATC cells (8505C) when compared to that of non-neoplastic controls [86]. However, silencing of *PVT1* resulted in a significant reduction of proliferation of ATC cells (8505C) followed by arresting cell cycle at G0/G1 stage. Cell cycle kinetics in ATC cells was changed due to PVT1 mediated decrease in cyclin D1 and thyroid-stimulating hormone receptor (TSHR) expressions [99]. Moreover, silencing of PVT1 halted recruitment of EZH2, which as a result ceased its enrichment. Furthermore, analysing the interaction between PVT1 and EZH2 by RNA immunoprecipitation assay in ATC cells indicated that PVT1 directly interacts with EZH2 to alter its expression in cells. Furthermore, the relative mRNA and protein levels of thyroid-stimulating hormone receptor in ATC cells (8505C) were significantly decreased after silencing of *PVT1* when compared to the controls [99]. Interaction of thyroid-stimulating hormone with its receptor plays a crucial role in the development and cell proliferation [103]. Thus, lncRNA PVT1 may contribute to tumourigenesis of ATC through recruiting EZH2 and regulating thyroid-stimulating hormone receptor expression.

Metastasis-associated lung adenocarcinoma transcript 1 (MALAT1) is a commonly overexpressed lncRNAs in human carcinomas [104,105]. *MALAT1* is associated with tumourigenesis, apoptosis, and cell cycle progression in various carcinomas, including ATC [106,107,108]. It acts as miRNA sponge by inhibiting the functions of tumour suppressor miR-363-3p, which in turn leads to Mcl1 expression, a target of miR-363-3p and allows Mcl1 mediated oncogenic role in ATC. Also, enhanced *MALAT 1* expression is closely associated with MEK/ERK signalling pathway [109,110,111]. MEK/ERK signalling reported to regulate cell cycle and apoptosis [109,110,111]. A study used MEK and Aurora kinase family inhibitor BI-847325 to determine its role in the expression of genes involved in cell cycle and apoptosis in ATC [108]. BI-847325 treatment significantly changed the expressions of *MALAT1* and its downstream target genes such as *miR-363-3p*, *Mcl1*, and cyclin D1 in ATC cells (C643 and SW1736)*. miR-363-3p* expression was significantly upregulated in both cells, whereas *Mcl1* and *MALAT1* expression were significantly down-regulated in C643 cells upon BI-847325 treatment [108]. Though *Mcl1* is a target gene of MEK/ERK signalling pathway, however, the expression of *Mcl1* was not significantly downregulated after BI-847325 treatment in the ATC (SW1736) cells [108]. Moreover, cyclin D1 is another oncogene affected by BI-847325 treatment in ATC cells as its expression was decreased followed by BI-847325 treatment [108]. Cyclin D1 is a prominent regulator of cell cycle and causes cell cycle arrest in G1 phase [112]. Thus, lncRNA MALAT1 mediated regulation of cell cycle and apoptosis related genes in ATC unveil potential target, which could be used as an effective option against ATC.

Another lncRNA known as papillary thyroid carcinoma susceptibility candidate 3 (PTCSC3) associated with ATC progression and doxorubicin resistance by targeting signal transducer and activator of transcription 3 (STAT3) protein [113,114]. STAT3 promotes tumourigenesis by enhancing inflammation, proliferation, invasion, and immunosuppression in cancer [113]. PTCSC3 was first identified in thyroid and plays tumour suppressive role in thyroid cancer [115,116]. STAT3 stimulates the expression of chromatin remodelling complex INO80, a member of INO80 subfamily, plays important functions in DNA replication, repair and transcriptional regulation [117]. PTCSC3 impedes the expression of INO80 through negatively regulating STAT3, which subsequently decreases drug resistance of ATC cells to chemotherapy drug doxorubicin [114]. In addition, PTCSC3 inhibits stem cell properties of ATC (8050C) cells, as the expression rate of CD133 and MDR-1 (cancer stem cell markers) was decreased [114]. Also, the expression of CD133 associated with poor disease outcome of patients with the cancer [114]. Furthermore, CD133 expressing ATC cells were involved in tumour progression, invasion and therapy resistance [118]. Thus, the lncRNA PTCSC3 regulates the pathogenesis of ATC by modulating STAT3 signalling and CSCs phenotype of cancer cells.

Additionally, non-coding pseudogenes such as high mobility group AT-hook 1pseudogene 6 and 7 (*HMGA1P6* and *HMGA1P7*) could regulate nuclear proteins such as high Mobility Group A (HMGA), which in turn regulate chromatin structure, replication and gene transcription, making them an important player in cellular processes [119,120]. Altered expression of HMGA has been associated with many pathophysiological diseases, including human carcinomas [119,120]. For example, higher HMGA1 protein levels are observed in ATC when compared to normal thyroid tissues [121]. Most importantly, *HMGA1P6* and *HMGA1P7*, two *HMGA1* non-coding pseudogenes, are abnormally expressed in ATC [121]. These non-coding pseudogenes bind with miRNAs such as miR-16 and miR-17, thereby release their inhibitory effects on target proteins such as HMGA1, which are involved in cancer progression [121]. Moreover, overexpression of *HMGA1P6* or *HMGA1P7* reduced the effects exerted by miRNA on the levels of both the *HMGA1* transcript and protein dose-dependently. Also, knockdown of *HMGA1P6* and *HMGA1P7* resulted in decreased *HMGA1* mRNA and protein levels, indicating that *HMGA1Ps* may compete for the endogenous miRNA-binding sites [121]. Furthermore, overexpression of *HMGA1Ps* lead to increase of HMGA2, and other cancer related proteins such as VEGF and EZH2 [121]. Thus, *HMGA1P6* and *HMGA1P7* are important proto-oncogenic competitive endogenous RNAs and could be used as a potential tool in designing anti-cancer therapy.

## 5. Non-coding RNA (siRNA) Mediated Targeted Therapy for Anaplastic Thyroid Carcinoma

Small interfering RNA (siRNA) or short interfering RNA, is a class of double-stranded non-coding RNA molecules operates within the RNA interference (RNAi) pathway (Figure 1) [122,123]. Exogenous siRNA mediates transcriptional gene silencing (TGS) in cells through DNA methylation and histone modification [124,125,126]. siRNA could silence EZH2 and then reverses the therapeutic drug (cisplatin)-resistance of non-small cell lung carcinoma and gastric carcinoma cells [127]. In ATC, gene silencing with siRNA has been implemented in cells proliferation, invasion, migration, metastasis, and apoptosis [99,101]. For example, siRNA mediated suppression of MADD (MAPK-activating death domain activating protein) expression inhibits growth and spread of ATC cells (8505C, C643 and HTH7) [128]. MADD could play a survival-promoting role by suppressing TNFα mediated apoptosis. MADD also activates MAPKs through Grb2 and Sos1/2 recruitment, which subsequently results in activation of ERK pathway without affecting the function of p38, Jun, and NFκB [129]. Accordingly, treatment with MADD siRNA inhibited in vitro proliferation of ATC cells (8505C, C643 and HTH7). Furthermore, it decreased in vivo tumour growth of ATC cell (8505C)-derived-orthotopic tumour model when compared to untreated control and scramble siRNA treated groups [128]. Furthermore, a significant reduction in cellular migration and invasion potential, as well as clonogenic capacity, were seen upon MADD inhibition. In addition, inhibition of (EMT) and Wnt signalling was observed after MADD inhibition [128]. Moreover, knockdown of MADD could exhibit its anti-migratory/invasive effects by blocking TNFα/ERK/GSK3β axis as MADD siRNA inhibited TNFα induced activation of pERK, pGSK3β and β-catenin [128]. Likewise, in vivo MADD siRNA treatment also decreased lung metastases along with tumour regression [128]. In addition, a reduction in proliferative index (Ki-67) and N-cadherin expression, and an increase in E-cadherin expression were observed in MADD siRNA-treated xenotransplanted tumour tissues [128]. Thus, MADD siRNA resulted in a broad range of effects in ATC cells such as inhibiting β-catenin nuclear translocation and consequently, the expression of its target genes.

siRNA mediated silencing of target genes associated with the induction of apoptosis of ATC cells [130]. For example, Nijmegen breakage syndrome 1 (NBS1) plays an important role in the repair of radiation-induced DNA double-strand breaks (DSBs) and suppression of NBS1 could induce apoptosis of ATC cells (8305C cells) [130]. The frequency of apoptosis through the caspase pathway and heat sensitivity was increased by the transfection of NBS1-siRNA in these ATC cells. Furthermore, NBS1-siRNA transfected cells were more frequent with histone H2AX phosphorylated at serine 139 (H2AX) foci than control cells transfected with scrambled siRNA [130]. H2AX foci are a kind of indicator for the presence of DNA double-strand breaks [131]. Specifically, one H2AX focus correlates with one DNA double-strand break [132]. DSBs have been found to be associated with heat shock-induced cell death [131]. Thus, NBS1-siRNA mediated heat sensitization could result from the suppression of heat-induced DNA double-strand breaks repair, which suggestedNBS1-siRNA could potentially function as a pro-apoptotic strategy for patients with ATC.

Inhibition of zinc finger transcription factor Slug by siRNA may induce apoptosis and sensitizes ATC cells (SW1736) to chemotherapeutic drug, doxorubicin [133]. Slug is a member of the Snail family of zinc-finger transcription factors, which could protect cells against apoptosis induced by radiation [134,135]. It mediates its action via regulation of Bcl-2 and Bax expression and transactivation of PUMA (p53 upregulated modulator of apoptosis) [136]. In addition, strong nuclear immunostaining of Slug was noted in ATC tissues and higher Slug mRNA expression was found in ATC cells (THJ-16T and THJ-21T) when compared to that of non-neoplastic cells [137]. Importantly, siRNA mediated inhibition of Slug inhibits the growth of ATC cells (SW1736) and sensitizes these cells to doxorubicin both in vitro and in vivo [133]. Moreover, siRNA mediated inhibition of Slug combined with doxorubicin resulted in reduced cell viability in ATC cells compared to the treatment with doxorubicin alone [133]. In addition, it was noted that silencing of Slug sensitizes ATC cells to doxorubicin via a PUMA dependent pathway. As inhibiting PUMA caused chemoresistance of ATC (SW1736)/Slug siRNA cells to doxorubicin [133]. On the other hand, the suppression of Slug in combination with doxorubicin significantly inhibited tumour growth and induced apoptosis in mice xenograft model. siRNA mediated suppression of PUMA reduced the threshold of apoptosis induction by DOX confirm the involvement of PUMA in apoptosis in ATC cells [133]. Therefore, knockdown of Slug has the potential for the development of new therapeutic strategies in order to improve chemotherapy for patients with ATC.

## 6. Conclusions and Future Perspectives

The patterns of expression of non-coding RNAs in ATC could be a useful tool for predicting prognosis of the patients with the cancer. Moreover, since non-coding RNAs are involved in the acquisition and enhancement of malignant properties by cancer cells, they could be potential targets for RNA-based therapy. Also, overexpression or inhibition of non-coding RNAs can lead to the reduction of the chemoresistance in ATC cells against some known anticancer agents. However, the difficulties in the analysis of this non-coding RNAs mediated regulatory network limit the investigation of their potential clinical applications. Thus, the improvements in gene and genome scanning technologies are needed to maximize the functions of non-coding RNAs in ATC. The utmost goal is to unveil their interactions in normal and cancer cells and finally to use this knowledge in designing anticancer therapies.

## Figures and Tables

**Figure 1 cancers-12-03159-f001:**
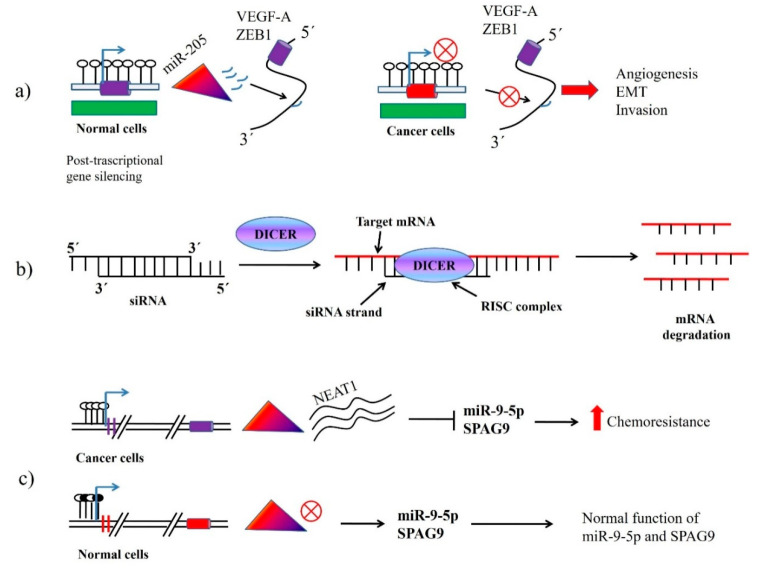
**Regulation of non-coding RNA mediated gene expression in anaplastic thyroid carcinoma (ATC)**. (**a**) In normal cells, miR-205can target and inhibit the expression of vascular endothelial growth factor A (VEGF-A) and Zinc Finger E-Box Binding Homeobox 1 (ZEB1). In cancer cells, the expression of miR-205 is downregulated, which subsequently results into angiogenesis, epithelial-mesenchymal transition (EMT) and invasion of ATC cells. (**b**) siRNA mediates gene silencing in ATC cells. siRNA interact with DICER and facilitate RNA-induced silencing complex (RISC) mediated mRNA degradation. (**c**) Nuclear enriched abundant transcript 1 (NEAT1) is upregulated in ATC cells where it inhibits the expression of miR-9-5p and Sperm associated antigen 9 (SPAG9), which increases chemoresistance of ATC. On the other hand, in normal cells, NEAT1 is downregulated, which frees miR-9-5p and SPAG9 to mediate their normal function.

**Figure 2 cancers-12-03159-f002:**
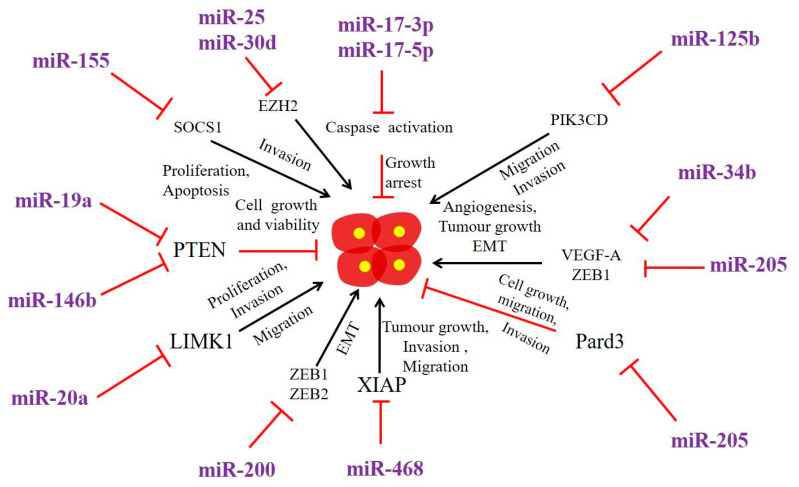
**Role of miRNAs in the regulation of gene expression in anaplastic thyroid carcinoma (ATC)**. miRNAs upregulate or downregulate the genes related to development, progression, invasion migration, and therapy resistance in ATC. PTEN: Phosphatase and tensin homolog; XIAP: X-linked inhibitor of apoptosis protein; SOCS1: Suppressor of Cytokine Signaling 1; PIK3CD: Phosphoinositide 3-kinase catalytic subunit delta.

**Table 1 cancers-12-03159-t001:** Potential functions of miRNAs in anaplastic thyroid carcinoma (ATC).

MiRNAs	Expression Pattern in ATC	Potential Target (s)	Potential Functions	Reference
miR-205	Downregulated	*VEGF-A*, *ZEB1*	Causes significant inhibition of angiogenesis and EMT Inhibits tumour growth, vascularization, migration and invasion.	[37]
miR-19a	Upregulated	*PTEN*	Increases cellular proliferation and progression by suppressing the expression of tumour suppressor *PTEN*	[38,39]
miR-17-3p and miR-17-5p	Upregulated	Unknown	Inhibition of these miRNA expression results in growth arrest and induction of apoptosis	[40]
miR-618	Downregulated	*XIAP*	Inhibits cell growth and induces apoptosis	[41]
miR-20a	Downregulated	*LIMK1*	Inhibits proliferation, migration and invasion	[42]
miR-155	Upregulated	*SOCS1*	Inhibits apoptosis, promotes proliferation, invasion and migration of ATC cells	[51]
miR-125b	Downregulated	*PIK3CD*	Decreases the PI3K, phospho-Akt, and phospho-mTOR expression in ATC cells Inhibits migration and invasion of ATC cells	[54]
miR-483	Upregulated	*Pard3*	Causes significant increase in cell growth, migration, and invasion.	[57]
miR-25 and miR-30d	Downregulated	*EZH2*	Inhibits proliferation and colony formation of ATC cells by inducing G2/M-phase cell-cycle arrest	[62]
MiR-34b	Downregulated	*VEGF-A*	Decreases cell proliferation, and wound healing, reduces cell cycle progression and increases apoptosis	[63]
miR-200 family	Downregulated	*ZEB1 and ZEB2*	Inhibits TGF-mediated EMT switch and decreases aggressiveness of ATC	[64]
miR-146b	Upregulated	*PTEN*	Inhibits the expression of tumour suppressor *PTEN* Decreases apoptosis, increases migration and invasion potential by regulating genes related to EMT Activates PI3K/AKT signalling to drive oncogenic proliferation	[67]

**Table 2 cancers-12-03159-t002:** Implications of lncRNAs in anaplastic thyroid carcinoma (ATC).

LncRNAs	Expression Pattern in ATC	Potential Target (s)	Potential Functions in ATC	Reference
H19	Upregulated	Unknown	Increases proliferation, migration and decreases apoptosis in ATC cells	[74]
HLA complex P5 (HCP5)	Upregulated	*miR-128-3p*	Increases cell viability, and decreases apoptosis	[78]
Urothelial carcinoma-associated 1 (UCA1)	Upregulated	*miR-135a*	Promotes ATC cell proliferation by acting as a competing endogenous RNA by binding with miR-135a Increases cell viability, proliferation, migration and invasion in ATC cells	[79]
Nuclear paraspeckle assembly transcript 1 (NEAT1)	Upregulated	*miR-9-5p*	Increases chemoresistance to cisplatin via miR-9-5p/SPAG9 axis in vitro and in vivo	[82]
Prader Willi/Angelman Region RNA5 (PAR5)	Downregulated	*Enhancer of Zeste Homolog 2* (*EZH2*) *E-cadherin*	Reduces cell proliferation and cell migration ability Increases E-cadherin expression Mediates its action by impairing EZH2 oncogenic activity	[90]
PVT1	Upregulated	*EZH2*	Increases cell proliferation by enhancing cell cycle progression, cyclin D and TSHR expression	[103]
MALAT1	Upregulated	*miR-363-3p*	Inhibits the expression of *miR-363-3p* by binding to it, thereby releases inhibitory effect of *miR-363-3p* to *Mcl1* oncogene Also decreases the expression of cell cycle and apoptosis related gene cyclin D1	[108]
PTCSC3	Downregulated	STAT3	Inhibits the expression of INO80 through negatively regulating STAT3, which subsequently decreases drug resistance of ATC to doxorubicin. PTCSC3 also inhibits stem cell properties of ATC 8050C cells	[114]
HMGA1Ps	Upregulated	*miR-16* and *miR-17* etc.	Leads to increase of HMGA2, and other cancer related proteins like VEGF and EZH2 levels	[121]

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
