# Peer review of "Roles of Non-Coding RNAs on Anaplastic Thyroid Carcinomas"

_cancers, 2020, doi:10.3390/cancers12113159_

Round 1

Reviewer 1 Report

In this review article, the authors discuss the roles of non-coding RNA in the main aggressive anaplastic thyroid cancer. This is a useful review on a new topic in cancer research, since non-coding RNAs could be used as target for this untreatable neoplasia.The review is well-written and balanced in its discussion, however, the section of long non-coding RNAs could be extended by adding the information of other lncRNAs present in literature (MALAT1 PMID: 31077090, PTCSC3 PMID: 29561707 and the HMGA1 pseudogenes PMID:25268743).

minor points:

please, in line 339 and 382, correct the name of non-coding RNAs and other typos.

Author Response

In this review article, the authors discuss the roles of non-coding RNA in the main aggressive anaplastic thyroid cancer. This is a useful review on a new topic in cancer research, since non-coding RNAs could be used as a target for this untreatable neoplasia. The review is well-written and balanced in its discussion, however, the section of long non-coding RNAs could be extended by adding the information of other lncRNAs present in literature (MALAT1 PMID: 31077090, PTCSC3 PMID: 29561707 and the HMGA1 pseudogenes PMID:25268743).

Response: Thanks a lot. We really appreciate your time and efforts in reviewing our manuscript. We have added the information from the literature as you suggested. It enriches our manuscript significantly. Please see lines 379-428.

Minor points: Please, in line 339 and 382, correct the name of non-coding RNAs and other typos.

Response: Thanks. All typos were corrected duly.

Reviewer 2 Report

Das et al. describe the role of non-coding RNAs in ATC in this review.  Overall, the content is on point, but there is some English grammar/sentence structure that needs to be edited and some writing that needs to be improved for clarity.

Comments/Questions:

  • Figure 1c siRNA seems out of place and should be with 1a combined with miRNA in some model that incorporates both or be Figure 1b
  • Line 96-97 doesn’t make sense “The tissue specific expression of antisense protein-coding genes of lncRNAs is similar to the expression pattern of lncRNAs,….” Not sure what the authors are trying to say.
  • Line 111-112: “due to some effective reasons” doesn’t sound right. Maybe “due to a variety of reasons.”
  • Lines 112-116: I’m not sure what the relevance of the miRNA locations is in the genome. Should clarify the intent of this comment.
  • I’m generally confused about the siRNA section because this review is on the role of non-coding RNAs in ATC which to me would mean focusing on miRNA and lncRNAs. The siRNAs are exogenous tools we use to study biology and not sure it fits in the current review.  I suppose one could combine siRNAs with miRNAs but note that DICER is operant in both
  • How do genetics relate to non-coding RNA expression patterns in ATC?

Author Response

Das et al. describe the role of non-coding RNAs in ATC in this review.  Overall, the content is on point, but there is some English grammar/sentence structure that needs to be edited and some writing that needs to be improved for clarity.

Response: Thanks a lot for your positive feedback. We have revised the manuscript extensively to improve the clarity and remove all the typos.

  • Figure 1c siRNA seems out of place and should be with 1a combined with miRNA in some model that incorporates both or be Figure 1b

Response: Thanks for this comment. We have modified the as suggested.

  • Line 96-97 doesn’t make sense “The tissue specific expression of antisense protein-coding genes of lncRNAs is similar to the expression pattern of lncRNAs,….” Not sure what the authors are trying to say.

Response:  Thanks. We have modified the section to clarify the context. Please lines 106-108.

  • Line 111-112: “due to some effective reasons” doesn’t sound right. Maybe “due to a variety of reasons.”

Response: Thank you. We have modified the lines 111-112 (now 120-121) as suggested.

  • Lines 112-116: I’m not sure what the relevance of the miRNA locations is in the genome. Should clarify the intent of this comment.

Response: Thanks. The intent is clarified by adding more information. Please see lines 124-125.

  • I’m generally confused about the siRNA section because this review is on the role of non-coding RNAs in ATC which to me would mean focusing on miRNA and lncRNAs. The siRNAs are exogenous tools we use to study biology and not sure it fits in the current review.  I suppose one could combine siRNAs with miRNAs but note that DICER is operant in both

Response: Thanks for this insightful comment. We have modified the subheading ‘Roles of small interfering RNA (siRNA) in anaplastic thyroid carcinoma’ to ‘Non-coding RNA (siRNA) mediated targeted therapy for anaplastic thyroid carcinoma’, as the whole section was about siRNA mediated intervention of ATC.

Our previous subtitle was misleading, now this section is discussing the targeted therapy for ATC using non-coding RNA siRNA as a model.

  • How do genetics relate to non-coding RNA expression patterns in ATC?

Response: The focus of the present study was to synthesize the epigenetic regulation of ATC, especially the roles of non-coding RNA in ATC. Also, the discussion of genetic and non-coding RNA is vast, thus, we did not study the relationship between genetic and non-coding RNAs in the present study.

Round 2

Reviewer 2 Report

The authors have done a reasonable job of make modifications.  English grammar and sentence structure needed.

Author Response

The authors have done a reasonable job of make modifications.  English grammar and sentence structure needed.

Response: Thank you again for revisiting our manuscript. We really appreciate your time and efforts. We have extensively revised the manuscript to remove the Grammar errors and to improve the structure.